# The Application of Microwaves, Ultrasounds, and Their Combination in the Synthesis of Nitrogen-Containing Bicyclic Heterocycles

**DOI:** 10.3390/ijms241310722

**Published:** 2023-06-27

**Authors:** Francesco Frecentese, Federica Sodano, Angela Corvino, Marica Erminia Schiano, Elisa Magli, Stefania Albrizio, Rosa Sparaco, Giorgia Andreozzi, Maria Nieddu, Maria Grazia Rimoli

**Affiliations:** 1Department of Pharmacy, “Federico II” University of Naples, 80131 Naples, Italy; federica.sodano@unina.it (F.S.); angela.corvino@unina.it (A.C.); maricaerminia.schiano@unina.it (M.E.S.); elisa.magli@unina.it (E.M.); salbrizi@unina.it (S.A.); rosa.sparaco@unina.it (R.S.); giorgia.andreozzi@unina.it (G.A.); 2Department of Medicine, Surgery and Pharmacy, University of Sassari, 07100 Sassari, Italy; marvi@uniss.it

**Keywords:** ultrasound, microwave, benzimidazole derivatives, indole derivatives, Imidazo[1,2-*a*]pyridines, Imidazo[[1,2-*b*]]pyridazines, Quinoline derivatives, isoquinoline derivatives, pyrrolopyridine derivatives, pyrrolopyridazine derivatives

## Abstract

The use of alternative energy sources, such as microwaves (MW) or ultrasounds (US), and their mutual cross-combination have been widely described in the literature in the development of new synthetic methodologies in organic and medicinal chemistry. In this review, our attention is focused on representative examples, reported in the literature in the year range 2013–2023 of selected N-containing bicyclic heterocycles, with the aim to highlight the advantages of microwave- and ultrasound-assisted organic synthesis.

## 1. Introduction

The term green chemistry can be traced back to 1991, but no publications mentioned this term in 1991 or 1992 [1]. Subsequently, the impetus for the development of this concept for obtaining, via eco-friendly methods, compounds of pharmaceutical interest, came from Anastas and Warner, who dictated 12 principles to be followed for this specific purpose [2]. The U.S. EPA (U.S. Environmental Protection Agency) has stated that “Green chemistry is the design of chemical products and processes that reduce or eliminate the use or generation of hazardous substances. Green chemistry applies across the life cycle of a chemical product, including its design, manufacture, use, and ultimate disposal. Green chemistry is also known as sustainable chemistry” [2].

In recent decades, great attention to green chemistry has been directed by both pharmaceutical companies and academia; this latter area has given strong impetus to the development of technologies and methods to ensure that drugs and lead compounds could be obtained in a manner compatible with the need to greatly reduce the environmental impact from the use of “traditional” methodologies, solvents, and reagents. However, greener synthetic routes to develop bioactive compounds, and the use of modern techniques such as ultrasound (US), microwaves (MW), flow chemistry and other approaches for the synthesis of biologically-active molecules, and safe transformations by degradable or recyclable reagents are still to be further developed, so that green chemistry can further provide its contribution to the quality of environmental matrices.

In tracing a path in the development of green chemistry, it is necessary to recall what were the first approaches developed. In particular, a first and simplest approach implemented by pharmaceutical industries has been to modify synthetic strategies for obtaining already known and marketed active ingredients using fewer toxic reagents and solvents, characterized by a reduced or no environmental impact [3].

Green chemistry principles and processes can decrease the negative impacts on the environment, but they cannot completely prevent the interaction of hazardous materials with the environment. The application of the principles of green chemistry, such as the use of appropriate solvents, specific catalysts, innovative technologies such as microwaves and ultrasound to carry out chemical reactions, and the choice of reagents that are less hazardous to human health unquestionably results in a reduction, but not a complete elimination, of problems related to the rate of pollution produced by the development of new, biologically-active molecules. In practice, green chemistry also results in a decrease in the expenditure of energy and materials, which obviously remain necessary to obtain the final compounds [4].

In this broad context, the aim of this review is to discuss the impact of microwaves, ultrasound, and their combined use in the synthesis of nitrogen-containing heterocyclic compounds, which are the most represented category of molecules of pharmaceutical interest, and numerous examples can be found in the categories of vitamins, anti-inflammatory drugs, anticancer drugs, antifungals, antibiotics, antivirals, etc. [5].

*N*-containing heterocyclic compounds dominate the field of biochemistry, medicinal chemistry, dyestuff, and photographic sciences and they are of increasing importance in many other areas including for use with polymers, adhesives, and molecular engineering [6].

In this review, our attention is focused on representative examples, reported in the literature in the year range 2013–2023, of N-containing bicyclic heterocycles, with the aim to highlight the advantages of microwave- and ultrasound-assisted organic synthesis. Our discussion is divided in sections relative to the following bicycles: indole, benzoimidazole, imidazopyridine, imidazopyridazine, quinoline, isoquinoline, pyrrolopyridine and pyrrolopyridazine. The advantages of applying MW and US for the synthesis of the bicyclic derivatives of our interest are illustrated in Figure 1a,b. In this wide context, the development of microwave flash heating and the application of ultrasounds, or their combined use, have proved to be particularly beneficial in the preparation of biologically-relevant N-containing bicycles, leading to very high yields, reduced reaction times, increased purities of intermediates, and, therefore, a dramatically lowered environmental impact [7,8,9,10].

### 1.1. The Use of Microwaves in Medicinal Chemistry

Among the technologies that have made wide inroads into the field of green chemistry, microwaves are certainly the ones that were first adopted by both industry and academia and they have, therefore, had the most widespread use.

The heating promoted by microwaves is based on the interaction of the electromagnetic field produced by instrumentation with molecules, resulting in the phenomenon known as “dielectric heating”. Microwaves are non-ionizing electromagnetic radiations characterized by a very low energy, which is not suitable for spontaneously promoting electronic transitions or chemical reactions [11]. Instead, electromagnetic radiation in the microwave range causes the rotation of the dipoles of molecules (having a non-zero dipole moment) as well as the acceleration of ions present in a sample. This phenomenon results in the orientation of the molecules toward the direction of the induced electromagnetic field and causes friction between the molecules themselves, thus leading to a rapid increase in the temperature within the sample itself. In more detail, the ability to generate heat results from the molecules’ ability to return to a “disordered” state. The heat generated in this way is considerably more homogeneous than in classical heating methods [11]. Microwaves pass through a vessel and lead to instantaneous localized heating. The radiation, being composed of an electric and magnetic field orthogonal to each other, induces a polarization of the molecule’s surroundings, which, by aligning with the generated field (ion dipole), generates “superheating” [12]. The kinetics of a reaction is related by the Arrhenius equation:k = Ae^−Ea/RT^
where A = Arrhenius constant, Ea = the activation energy, R = the gas constant, and T = the absolute temperature (K).

Microwaves are electromagnetic waves that lie in the region of the spectrum between infrared radiation and what are usually called radio frequencies. The characteristic wavelengths of microwaves are between 1 cm and 1 m, and the frequencies of 30 GHz and 300 MHz correspond to them, respectively. The region of the spectrum between 1 cm and 25 cm is usually used in civilian and military RADARs, it is essentially so as not to interfere with these, therefore, that domestic and industrial ovens operate at the very specific frequencies of 2450 GHz and 900 MHz (i.e., wavelengths of 12.2 cm and 33.3 cm, respectively) [13]. Domestic microwave ovens generally operate at frequencies of 2.450 GHz.

To date, the days of the first “pioneering” experiments conducted in household-type microwave ovens, where safe conditions and the monitoring of the temperature, pressure and power were by no means guaranteed, are long gone.

The instrumentation available today is specifically designed for organic synthesis and consists of ovens operating at the frequency of 2.45 GHz; they in turn are distinguished into two categories defined as monomode MW, which is specifically intended for small-scale reactions conducted in closed vessels (0.2 to 50 mL; pressure under 20 bars), small open vessels or solvent free [8], and multimode MW, where the microwaves do not directly hit the sample but rather bounce against the oven walls, which thus constitute a “resonant cavity” and, therefore, become suitable for larger samples [8].

The polarity of a solvent plays a determining role on its ability to absorb microwaves. The factors that contribute to the individual absorption of solvents are mainly the dielectric constant, dipole moment and dielectric loss tangent (tan δ). In particular, the ability of a material or substance to convert electromagnetic energy into heat at a given frequency and temperature is determined by the following equation:Tan δ = ε″/ε′

Tangent delta (δ) is the sample dispersion factor or the ability of a material to absorb microwave energy by converting it into heat energy. It is defined as the ratio of the dielectric loss (i.e., ε″, that measures the efficiency with which the energy of electromagnetic radiation is converted into heat) and the dielectric constant (i.e., ε, the degree or tendency of a molecule to be polarized by an electric field) [13].

When setting experiments, chemists have to consider, therefore, the dielectric properties of a reaction mixture and they should consider the pressure generated by a particular solvent in the reaction vessel at certain temperatures.

When solvents, due to their toxicity or to their dielectric properties, are not suitable for the desired chemical transformation, they can be replaced by ionic liquids as useful substitutes for common organic solvents, because they are less toxic and have unique chemical and physical properties. They are salts with melting points at or near room temperature and depending on the cation or anion, they can be basic, neutral, or acidic; moreover, they can be hydrophilic or completely hydrophobic; and finally, for very wide ranges of temperatures (i.e., even more than 300 °C), they are in the liquid state and have a very low vapor pressure [14].

The availability of the latest generation of reactors dedicated to microwave-assisted organic synthesis has made it possible to carry out chemical reactions at temperatures above the boiling point of solvents, in a temperature range that now allows for over 300 °C and 200 bar pressure. Reactions can be conducted in an inert atmosphere, in the absence of solvents [15] or by making use of ionic liquids [16] that promote the absorption of electromagnetic radiation. In addition, the availability of rotating carousels makes it possible to run multiple reactions simultaneously, facilitating the obtaining of libraries of compounds to be submitted to biological screening steps; thus, greatly accelerating the drug discovery process.

Finally, the attention of medicinal chemists in recent years has been particularly focused on scale-up steps to enable large quantities of biologically-active compounds to be obtained in a reduced time. Such studies have enabled the development of synthetic protocols based on the use of continuous flow reactors, which are available for both types of microwave instruments, thus ensuring the possibility, especially for pharmaceutical industries, of producing kilograms or even larger quantities of actives through such green technology [17].

### 1.2. Ultrasounds in Medicinal Chemistry

The term “sonochemistry” is defined as an interdisciplinary branch of chemistry focused on the study of the effects of acoustic waves on chemical systems. Applications of ultrasound, i.e., waves capable of generating phenomena such as sonoluminescence, sonic cavitation, sonolysis and sonocatalysis, are well reported in the literature [18]. These different phenomena are the reason for the versatility of ultrasound.

Ultrasounds are sound waves with frequencies above 20 kHz (i.e., the limiting frequency of sounds audible in humans). They are generally classified into low-frequency, intermediate-frequency and high-frequency ultrasound based on the several physical mechanisms that can be induced. Consequently, depending on the frequency they have, they have different uses ranging from diagnostic and therapeutic purposes to industrial applications (i.e., low frequency ultrasounds).

Most applications of low-frequency ultrasound rely on cavitation effects. Cavitation is a phenomenon consisting of the formation of vapor zones within a fluid (cavitation microbubbles) that then implode, producing a characteristic noise. When ultrasound is applied in a liquid–liquid or solid–liquid system, cavitation microbubbles are generated from the gas nuclei produced by the liquid or trapped in the reactor wall. As the pressure increases, these microbubbles, which have a stable shape and volume, are subjected to ultrasonic-induced stresses and then collapse, creating liquid microcurrents, jets and shock waves. This severe collapse gives rise to very high temperatures and local pressure variations, which produce intermediate ion radicals that can react with reagents and, thus, accelerate some reactions. The intensity of the stresses will depend on the frequency of the ultrasound and the size of the microbubbles [19,20]. When these bubbles implode, they release a large amount of energy and induce, in a localized form, extreme conditions of temperature (up to 5000 K) and pressure (up to 1000 bar), leading to reactions involving high-energy free radicals and physical effects, thereby making it possible to perform heterogeneous phase reactions, in water, without solvents, and allowing for cleaner products with less irradiation time [21].

There are different types of sonochemical reactors, commonly called “sonicators”; they mainly consist of a generator that converts electrical energy into ultrasound at 20 KHz, a transducer that converts this energy into a longitudinal mechanical vibration of the same frequency, and a probe that increases the amplitude of the vibrations to be transmitted to the products to be processed. Different types of sonicators differ in terms of their basic configuration, geometry, number of transducers, and position of transducers.

Based on these parameters, we essentially distinguish between two types of sonicators: an immersion sonicator and an ultrasonic bath.

The immersion sonicator operates at a fixed frequency, but it is possible to control the cavitational intensity in the reactor by varying the amplitude. It consists of a long ultrasonic cone, being either horizontal or vertical, that acts as a transducer and that must be immersed directly into a liquid. The cavitation activity is highest near the transducer.

The ultrasonic bath, on the other hand, consists of a tank of varying sizes and depending on its capacity there may be one or more transducers inside it. The flexibility of their number and, more importantly, their location allows for a homogeneous distribution of cavitational activity in the reactor. In addition, irradiation can be performed directly or indirectly. In the former case, the liquid is taken directly into the bath, while in the latter case the liquid is taken from a reactor and then immersed in the ultrasonic bath, using a coupling fluid. The indirect mode, however, implies a restriction on the operating conditions, particularly on the temperature, as this cannot exceed the boiling point of the coupling fluid, which, most often, is water. For large-scale operations, continuous-flow cell sonicators are preferred in which the transducers are attached to the wall of a reactor, which may be hexagonal or rectangular in shape. In this way, a uniform distribution of the cavitational activity such as that of an ultrasonic bath can be achieved. In addition, because of the lower power dissipation per unit transducer, decoupling losses can be reduced, which means that more energy will be available for the desired transformations [20].

### 1.3. Combined Use of Microwaves and Sonochemistry

The idea of the possible use of a combination of microwave flash heating and sonochemistry arose in 1999 [22] and had a great impact among chemists. As for the single techniques, the first approaches were based on “inhouse” instruments.

These early experiments were carried out notably by Lionelli and Mason [23], who tried different combinations related to the simultaneous use of microwaves and ultrasound to accelerate and improve the yield in chemical reactions.

Examples based on two different ways of coupling the energy sources were reported in a paper published by these authors: one involving the use of a pump, which in a continuous flow allowed the reaction mixture to pass through separate reactors, each destined for a radiation source; the other in which both microwaves and sound waves were sent within the same reactor.

To date, both of the two modes have become commercially available. At the same time, today, these instruments have also made it possible to significantly reduce the time for digesting and dissolving samples for analytical purposes [24].

In the years between 2000 and 2010, a great impetus for the development of the combined use of the two green technologies for performing chemical reactions was provided by the research group of Cravotto et al. [19,25,26,27].

In their papers, it is described in detail how it was possible to overcome some specific issues related to the coupling of the two techniques. For example, the transducers for the generation of ultrasound was generally made of metallic material and this did not match with the possibility of inserting them in the cavity of microwave ovens. For this problem, the resolution was the realization of transducers that were made of Pyrex or quartz, with the latter material, however, not being the preferred one because of its high fragility. The immediate solution to this problem, of course, was the availability of ultrasonic transducers made of plastic material such as polytetrafluoroethylene (PTFE), that was remarkably resistant to thermal and mechanical shock. The availability of microwave ovens specifically intended for organic synthesis allowed the combined use of MW and US to progress as was already reported in the section on MW; but domestic microwave ovens had a lack of probes for determining the internal temperature of a sample, the temperature of a vessel, the pressure of any solvents placed inside a reactor, and the microwave power that was used for heating. When professional instruments became commercially available, the combined use of the two green techniques obviously also benefited from these improvements.

## 2. Indole

Bicyclic heterocyclic structures are frequently detected in both natural and synthetic compounds of biological significance, with indole being the most prevalent scaffold of this kind. Dozens of synthetic routes have been developed to afford the creation of thousands of compounds based on the indole scaffold [28,29].

The indole scaffold plays a crucial role in various industrial sectors, including pharmaceuticals, food and food supplements, dyes, and paints.

In the pharmaceutical field, indole has proven to be a highly favored and versatile scaffold for the synthesis of biologically-active molecules with diverse pharmacological activities, ranging from being anti-inflammatory and antineoplastic to antibacterial, antitubercular, anti-hepatitis, and more [30].

Due to the significance of the indole nucleus in the discovery of new lead compounds, numerous examples detailing the synthesis of compounds incorporating this scaffold using microwave and ultrasound techniques have been documented in the literature. These reports contribute to expanding the range of the compound libraries that can be obtained through increasingly efficient and environmentally friendly synthetic methods.

### 2.1. Microwave Assisted Synthesis of Indole Derivatives

In a recently published paper, Bellavita et al. [31] reported an effective procedure to synthesize a series of functionalized 2-methyl-1H-indole-3-carboxylate derivatives (**1**–**8**) from commercially available anilines suitably functionalized by different electron-withdrawing and -donating groups through a catalyzed palladium.

N-aryl enamine carboxylates with different electron-withdrawing (-NO_2_, -Cl, -Br) or -donating groups (-CH_3_, -OPh) (Figure 1) were synthesized and then converted into their corresponding indoles under both MW-assisted and conventional heating conditions by palladium-catalyzed oxidation.

The synthesis of indoles **1**–**8** was carried out separately in different solvents as dimethylformamide (DMF), dimethyl sulfoxide (DMSO) and acetonitrile (ACN) to investigate the impact of the solvent on the reaction rate. Among the solvents tested, the DMF exhibited the most favorable results for the majority of the substituted *N*-aryl enamine.

A decrease on the reaction time, enhanced conversions, and cleaner product formation were reported in the proposed strategy that involves the use of four players: (i) a copper source, (ii) a ligand, (iii) a base, and (iv) an N-aryl enamine, to efficiently obtain a series of 2-methyl-1H-indole-3-carboxylate derivatives.

In another paper by Rathod et al. [32], a valuable microwave-assisted synthesis of isoniazid and indole derivatives has been reported aiming to obtain new efficient antitubercular agents. In this paper, the Betti’s multicomponent reaction, based on the condensation of an aldehyde, an amine and phenol was explored, as a one-pot/solvent-free procedure, to access a small library of novel antitubercular agents (Figure 2, compounds **9a**–**d**, **10a**–**d** and **11a**–**d**).

In vitro antitubercular assays demonstrated that compounds **9a** and **9c** exhibited minimal inhibitory concentration (MIC) values comparable to those of streptomycin. Molecular docking studies were conducted for these two lead compounds, revealing a binding mode similar to that of the parent drugs. This finding enhances the potential for considering them as new anti-resistant tubercular drugs.

A green one-pot procedure for the preparation of 3-functionalized indoles has been reported in the literature by Wei Lin et al. that described a microwave-assisted regioselective domino reaction [33]. The synthetic route provided the indoles **12** by a reaction of anilines, arylglyoxal monohydrates, and cyclic 1,3-dicarbonyl derivatives without the formation of the corresponding hydroquinolines (Figure 3). The reaction was performed by microwave heating using a mixture of ethanol and water. Optimized conditions for the reaction were found to be the following: trifluoroacetic acid (TFA) as a catalyst, and MW (at 90 °C, for 40 min). The yields were determined by HPLC–MS on the crude reaction mixtures.

### 2.2. Ultrasound-Assisted Synthesis of Indole Derivatives

Damavandi et al. [34] successfully synthesized novel 2-amino-4,5-dihydro-4-arylpyrano[3,2-b]indole-3-carbonitriles **13** with the aid of ultrasonic irradiation and potassium dihydrogen phosphate (KH_2_PO_4_) as the catalyst. The reaction was carried out via the three-component condensation of indoles, aromatic aldehydes and malononitrile in ethanol at 60 °C (Figure 4).

Dandia et al. [35] described a simple and ultrasound-assisted synthesis of spiro[indole-3,4′-pyrazolo[3,4-*e*][1,4]thiazepines] **14**. Indeed, these compounds were obtained by one-pot, three-component domino reactions of isatins, 3-amino-5-methylpyrazole and 2-mercaptoacetic acid derivatives, catalyst-free and in aqueous media, to afford a corresponding biologically-relevant excellent yield at room temperature and in a shorter time frame (Figure 5). The adopted protocol allowed only the new seven-member ring system to be selectively obtained, as confirmed by a single-crystal X-ray analysis of a representative compound and spectroscopic techniques. Compared with conventional synthesis, the advantages of this method are the absence of catalysts, ease of processing, and the use of water as a solvent, which is environmentally friendly; in addition, this work offers a new way to create molecular complexity with ease. Representative compounds were investigated to assess their anti-hyperglycemic activity in terms of their capability to inhibit the enzyme α-amylase. Ultrasonic irradiation was found to be superior as compared with the traditional methods with respect to the reaction time and yields. Several reaction conditions were screened for compound **14a**, as shown in Table 1.

In a recent paper, Vieira et al. [36] described a new atom-economic and selective method to make 3-selanylindoles **15** through the direct selanylation of indole derivatives (Figure 6). By using this US-promoted reaction, with CuI (20 mol%) as the catalyst and DMSO as the solvent, eleven 3-organylselanylindoles were prepared selectively in higher yields and shorter reaction times. The reaction conditions were optimized and those for compound **15a** are shown in Table 2. The advantage in using US over the other heating systems (i.e., conventional heating or microwave irradiations) was observed by carrying out a comparative study.

Jaiswal et al. [37], for the first time, described an efficient, simple, synthetic green protocol for the one-pot synthesis of functionalized 2-oxo-benzo[1,4]oxazines **16** in water using ultrasound irradiation (Figure 7). As compared to standard methods, the adopted protocol led to a reduced reaction time and furnished the target molecules in excellent yields (up to 98%) with no side products. For the first time, this protocol was productively applied to the synthesis of cephalandole **16a**, an indole antitumor alkaloid.

Pogaku et al. developed a novel and green multicomponent protocol for the synthesis of a new series of potent spirooxindolopyrrolizidine-based 1,2,4-triazole-1-yl-pyrazole antitubercular derivatives using the ionic liquid 1-butyl-3-methyl imidazolium tetrafluoroborate) [B_mim_]BF_4_ by the ultrasonic method [38]. This synthetic method (see Figure 8) had significant advantages, such as excellent yields in a reduced reaction time and product isolation, a simplicity of separation, and the recyclability of [B_mim_]BF_4_.

## 3. Benzimidazole

The benzimidazole core is found in a variety of pharmaceuticals; therefore, its derivatives are being researched in medicinal chemistry because of their pharmacological interest. Benzimidazole shows antibacterial [39], anti-inflammatory [40], antiviral [41,42], and antitumor activities [43]. In view of their widespread pharmacological activity, numerous procedures have been applied for the synthesis of benzimidazole and its derivatives.

### 3.1. Microwave-Assisted Synthesis of Benzimidazole Derivatives

The first attempt for the microwave-assisted preparation of benzimidazole appeared in 1995 and the report described the condensation of 4-substituted-1,2-diaminobenzene with ethyl acetoacetate or ethyl benzoylacetate under microwave irradiation in a domestic oven [44]. Since then, dozens of papers have reported various procedures for the MW-assisted preparation of benzimidazole derivatives.

Examples of scientific papers that have appeared in the literature in recent years include one related to the multicomponent Ugi reaction published by Song et al. in 2016 [45]: amino pyridine was reacted in a three component reaction with Boc-protected isonitrile and an aromatic aldehyde in methanol at room temperature, affording the corresponding imidazo[1,2-*a*]pyridine in appreciable amounts. The following deprotection and cyclization reaction was then optimized by microwave heating to give the fused benzimidazole system **18**. The optimized reaction conditions were settled to be 5% HCl/AcOH as the solvent, at 180 °C for 30 min under microwave irradiation. Adopting this procedure, six different examples (named **18a**–**18f**, Figure 9) of benzimidazole derivatives were prepared in a yield range of 60–72%.

Giving due consideration to the fact that the versatile activity of the benzimidazole heterocyclic nucleus has allowed the commercialization of anti-neoplastic, anthelmintic, anti-fungal, and anti-ulcer drugs, much attention in recent years has been placed on the effects of benzimidazole derivatives on certain enzymes, such as topoisomerases and protein kinases [46]. In this context, besides the neutral benzimidazole derivatives, N, N-disubstituted benzimidazolium salts have gained attraction and have been described as human carbonic anhydrase I and II inhibitors.

Twelve benzimidazolium salts (**19**) were prepared as described in Figure 10 by reacting p-toluenesulfonic acid and 1-substituted-benzimidazole in ethanol under 200 W of microwave irradiation for 30 min. Two of these compounds were further converted into iodide salts by a treatment with NaI in ethanol at room temperature for 24 h (with no microwave).

All the compounds reported in this study showed a lower inhibition level in comparison to sulfonamides and other reported types of carbonic anhydrase inhibitors, but at the same time, the compounds revealed a very high solubility and a noncompetitive mechanism of interaction which is uncommon in this class of inhibitors.

In 2022 a Malaysian research group reported the microwave-assisted N-alkylation of the benzimidazole scaffold for the preparation of new N-alkyl-substituted benzimidazolium salts acting as anticancer and antibacterial agents (Figure 11) [47].

The duration of N-alkylation yielding **20a**–**f** benzimidazoles resulted in being shortened from 2 days to 1 h, with approximately a 20% increment in the yield.

The most promising compounds of this series were **20e** and **20f** for the antibacterial and anticancer profiles, respectively; in both cases the activity was correlated to the high alkyl chain length.

In the same year, Nardi et al. published a very useful microwave-assisted method for obtaining 1,2-disubstituted benzimidazoles **21** via 1% mol of erbium(III) triflate Er(OTf)_3_ catalysis (14 examples—Figure 12) [48]. The synthetic procedure is fast (i.e., from 5 to 12 min) and is very high-yielding (i.e., between 86 and 99%). The method is mild and allows access to a variety of benzimidazole derivatives in solvent-free conditions. The synthetic procedure not only was proved to be environmentally friendly but also resulted in the isolation of all benzimidazole derivatives by adding water to the crude mixture and then extracting with ethyl acetate. The reaction procedure resulted in being efficient also in a large scale synthesis, proving the potential industrial scalability of the method.

### 3.2. Ultrasound Assisted Synthesis of Benzimidazole Derivatives

Karami et al. [49] have developed a new polymer—supported by trifluoromethanesulfonic—for the synthesis of benzimidazoles **22** using aldehydes and [1,2-*b*]enzenediamine irradiated by ultrasounds under solvent-free conditions in the presence of a catalytic amount of poly (4-vinylpyridine), supported with trifluoromethanesulfonic acid (PVP-TfOH). This method has several advantages including short reaction times, mild conditions, excellent yields, it is inexpensive, and it has a non-toxic catalyst, a simple operation and work-up. The elimination of the solvent has obvious environmental benefits, and the catalyst could be successfully recovered and re-used for at least three runs without a significant loss in activity. An example is reported in Figure 13.

Nile et al. [50] have developed a speedy, clean and chemo-selective process under ultrasound irradiation at 50 °C for the one-pot synthesis of a series of benzimidazoles using Amberlite IR-120 in an EtOH/H_2_O mixture. The condensation reaction resulted in the desired products with considerably good yields of 2-aryl-1-arylmethyl-1H-benzimidazole derivatives **23** (Figure 14).

Chen et al. [51] have disclosed a simple, ultrasound-assisted piperidine-catalyzed three-component protocol for the synthesis of functionalized benzimidazo[2,1-b]quinazolin-1(1H)-ones **24**. Substituted 2-aminobenzimidazoles, aromatic aldehydes and 1,3-cyclohexadiones were reacted in aqueous isopropyl alcohol (IPA) at room temperature (Figure 15).

Naeimi and Babaei [52] have developed a new methodology for the synthesis of 2-aryl benzimidazoles **25**. The model reaction involved *o*-phenylenediamine and aryl aldehyde derivatives using MnO_2_ nanoparticles as a convenient oxidant agent in ethanol–water (1:1) as the solvent under ultrasound irradiation (Figure 16). In this protocol the desired products were purely obtained in high yields with an easy work-up and short reaction times.

Ziarati et al. [53] described in their work how various aldehydes reacted with benzene-1,2-diamine in the presence of rare-earth nanostructured NiFe_2_–xEuxO_4_ as a catalyst in water under ultrasound irradiation, successfully giving benzimidazole derivatives **26** (Figure 17). According to Table 3 and Table 4, several conditions were used for the preparation of benzimidazoles **26** and with different methods (i.e., conventional versus sonication); however, the best condition was under ultrasonic irradiation.

Sapkal et al. [54] have developed a practical and convenient synthetic method in aqueous media by using bismuth oxychloride (BiOCl) nanoparticles for the synthesis of 2-substituted-1*H*-benzimidazoles **27** under ultrasound irradiation. The simple methodology with short reaction times and mild reaction conditions, an easy processing procedure and ease of recovery, and reusability of the catalyst, BiOCl NPs are the salient features of this method (see Table 5).

## 4. Imidazo[1,2-*a*]pyridines

Imidazo[1,2-*a*]pyridines are important building blocks very common in many biologically-active compounds and natural products. Prominent among them are imidazo-[1,2-*a*]pyridines, which are of great pharmaceutical interest as they exhibit a wide range of pharmacological activities. Many commercially available drugs, such as zolimidine (for peptic ulcer), zolpidem (for insomnia), and alpidem (anxiolytic agents), contain imidazo[1,2-*a*]pyridine moiety. Due to their noteworthy biological and synthetic value, the construction of imidazo[1,2-*a*]pyridine scaffolds has been a prominent research topic, and significant progress had been made in the past decades [55].

### 4.1. Microwave-Assisted Synthesis of Imidazo[1,2-a]pyridine Derivatives

Heibel et al. [56] have developed an efficient method to obtain various 2,3-diarylimidazo[1,2-*a*]pyridines **28** in polyethylene glycol-400 (PEG400). After optimization, a useful one-pot process enabled access to 2,3-diarylimidazo[1,2-*a*]pyridines by using a reduced amount of palladium catalyst. PEG400 is described herein as a suitable medium for the condensation of various 2-aminopyridines with α-bromoketones. Here, 2-Arylimidazo[1,2-*a*]pyridines were obtained in a brief time frame through microwave irradiation with moderate to excellent yields (Table 6).

Wagare et al. [57] developed a useful protocol for the synthesis of 2-phenylimidazo[1,2-*a*]pyridines **29** from cyclo-condensation of in situ-generated α-bromoacetophenones and 2-aminopyridines in PEG-400 and water (1:2), under microwave irradiation (Figure 18). The procedure provided a better alternative to the current method as it avoided the use of lachrymatric α-haloketones as well as volatile toxic organic solvents, and it cut down the reaction time to obtain imidazo[1,2-*a*]pyridines in a relatively high yield.

Karamthulla et al. [58] elaborated an iodine-mediated one-pot, three component reaction for the synthesis of 2,3-disubstituted imidazo[1,2-*a*]pyridines **30**, by using a 1,3-binucleophilic nature of 2-aminopyridines under microwave irradiation (Figure 19). A wide variety of compounds could be obtained in good to very good yields using this methodology and all the reactions took place within brief times and were metal-free.

Lee et al. [59] examined the microwave-assisted palladium-catalyzed Suzuki reaction coupling of 3,5,6, and 7-bromoimidazo[1,2-*a*]pyridine with arylboronic acids that provided diversified arylimidazo[1,2-*a*]pyridine derivatives **31** with a short reaction time using a 1 mol % of Tetrakis(triphenylphosphine)-palladium(0), Pd(PPh_3_)_4_, and an equimolar concentration of caesium carbonate (Cs_2_CO_3_) in DMF at 130 °C for 40 min (Figure 20).

Rao et al. [60] developed a catalyst-free heteroannulation for imidazo[1,2-*a*]pyridines **32** starting from substituted 2-aminopyridines and α-bromoketones under microwave irradiation and using water–isopropyl alcohol (H_2_O-IPA) as the reaction medium (Figure 21). All the synthetic compounds, obtained in excellent yields and with brief reaction times in a single synthetic operation, were studied for their in vitro anti-inflammatory and antimicrobial activities. The anti-inflammatory data indicated that the compounds with a 2-nitro or 4-methyl substituent on the imidazo[1,2-*a*]pyridine moiety showed significant albumin-denaturing power, and in addition, some of the tested compounds demonstrated antimicrobial activity by inhibiting the growth of Proteus and Klebsiella bacteria.

Hernandez et al. [61] developed a useful microwave-assisted Groebke–Blackburn–Bienaymé reaction (GBBR) protocol for the eco-friendly synthesis of imidazo[1,2,*a*]pyridine-chromones **33**. In this strategy, 3-formylchromone, 2-aminopyridine, and several substituted isocyanides were reacted with a 20 mol % ammonium chloride catalyst in EtOH under MW-irradiations at 80 °C for 15 min, which afforded the chromones-imidazo[1,2-*a*]pyridine hybrid **33** in a 21–36% yield range (Figure 22). Chromones and imidazo[1,2-*a*]pyridines are privileged cores of high interest in medicinal chemistry.

Rodríguez et al. [62] synthesized the 2-phenylimidazo[1,2-*a*]pyridines **34** via the simplest route taking 2-aminopyridine and substituted α-bromoketones in the presence of sodium bicarbonate in methanol MeOH at 80 °C for 1 min under MW-irradiations in a 24–99% yield range. It was also found that the improved yields in the present methodology was observed as compared to other, more tedious, methodologies such as thermally- and mechanically-assisted routes (Figure 23).

### 4.2. Ultrasound-Assisted Synthesis of Imidazo[1,2-a]pyridine Derivatives

Kurva et al. [63] developed the first efficient and mildly USI-assisted Groebke–Blackburn–Bienaymé reaction (GBBR) to obtain new carbazolyl imidazo[1,2-*a*]pyridine-3-amines **35** of the bonded type, employing ammonium chloride (10 mol%) as a catalyst (Figure 24). The adopted methodology has several advantages, such as the ability to synthesize bonded-type bis-heterocycles containing two preferred scaffolds in a single step, a wide tolerance of functional groups, an ease of reaction, short reaction times, operational simplicity, excellent yields, it is environmentally friendly and has inexpensive solvents and catalysts.

Vieira et al. [64] synthesized a series of imidazo[1,2-*a*]pyridines **36** via an ultrasound-mediated reaction of 2-aminopyridine with 2-bromoacetophenone derivatives. This protocol tolerated a wide range of 2-bromoacetophenone derivatives to produce a variety of imidazo[1,2-*a*]pyridines 36 in good to excellent yields. This strategy was efficiently extended to the one-pot synthesis of 3-selanylimidazo[1,2-*a*]pyridines **37** by the reaction of imidazo[1,2-*a*]pyridines formed in situ with diorganyl diselenide in the presence of copper(II) sulfate/potassium iodide under sonication (Figure 25). All the reactions were conducted in an air atmosphere using PEG-400 as a non-toxic solvent compatible with the ultrasound conditions in an environmentally-benign process.

Paenghua et al. [65] reported a convenient and efficient two-step ultrasonic-assisted process for the synthesis of substituted imidazo[1,2-*a*]pyridines **38** using acetophenone and 2-aminopyridine in the presence of iodine and 1-butyl-3-methylimidazolium tetrafluoroborate ([B_mIm_]BF_4_) as a catalyst, followed using cesium fluoride CsF-Celite as a solid base (Figure 26). The use of CsF-Celite in the post-treatment, instead of the use of aqueous bases of NaOH and K_2_CO_3_, offered various features such as an operational simplicity in terms of phase separation, a reduced post-treatment time, while it generated the expected products of **38** in good yields.

Yang et al. [66] reported an ideal strategy for preparing the regioselective iodination of imidazo[1,2-*a*]pyridines **39** at their C3 position. The reaction advanced through ultrasound acceleration in the presence of an ecofriendly, alcohol-based solvent and was promoted by tertbutyl hydroperoxide (TBHP) without any additive, base, metal, or an inorganic oxidant (Figure 27). This reaction featured good functional-group tolerance, a high selectivity, was transition-metal free, and could occur under mild conditions.

Ramarao et al. [67] developed a rapid and environmentally friendly sonochemical approach to synthesize the imidazo[1,2-α]pyridine derivatives **40**. Their synthesis was carried out by an NBS (*N*-bromosuccinimide)-promoted reaction of 2-aminopyridines with β-keto esters (or 1,3-dione derivatives) in PEG-400 under ultrasonic irradiation (Figure 28). This sonochemical method yielded the desired product in a good yield when 2-aminopyridines containing an electron-donor methyl group were used, while the corresponding products were obtained in a low yield when an electron-withdrawing chlorine group was present in the 2-aminopyridines. Additionally, the mechanism of reaction is reported in Figure 28.

## 5. Imidazo[1,2-*b*]pyridazine

The imidazo[1,2-*b*]pyridazine core, which is an imidazole ring fused with a pyridazine moiety with a nitrogen atom at the ring junction, represents an important class of the imidazopyridazine family. In fact, the core imidazo[[1,2-*b*]]pyridazine confers wide applications in medicinal chemistry as it is anticancerous, antiparasitic, antimicrobial, antiviral, antidiabetic, and antineuropathic. In addition, imidazo[1,2-*b*]pyridazines are well known as antimycobacterial agents with locomotor activity in vivo and have been used as ligands for Alzheimer-type β-amyloid plaques [68].

### Microwave-Assisted Synthesis of Imidazo[[1,2-b]]pyridazine Derivatives

Behbehani and Ibrahim [69] reported the first example of a microwave-assisted one-pot synthesis protocol for benzo[4,5]-imidazo[1,2-*b*]pyridazines **41** using water as a green solvent in the presence of sodium acetate (Figure 29). The whole strategy consists of just one step, namely, a reaction between 3-oxo-2-arylhydrazonopropanals, which contain an *o*-fluorine substituent on the *N*-aryl ring of the arylhydrazone moieties with active methylene compounds, including 3-oxo-3-hetarylpropionitrile, 3-oxo-3-phenylpropionitrile, ethyl cyanoacetate and 2-cyanoacetamide, giving the target compounds **41** in a global yield of 89–99%.

Moine et al. [70] developed a new method for the double functionalization of imidazo[1,2-*b*]pyridazines with highly potent anti-toxoplasma gondii activity, targeting TgCDPK1. As depicted in Figure 30, 2,3-diarylimidazo[1,2-*b*]pyridazines were obtained starting from imidazo[1,2-*b*]pyridazin-2-yl-triflate after a treatment with boronic acid under the following Suzuki–Miyaura reaction conditions (i.e., ArB(OH)_2_, Pd(PPh_3_)_4_, Na_2_CO_3_, and 1,4-dioxane/H_2_O (2:1), at 100 °C, MW, for 30 min). Using this protocol, the intermediate was isolated in an 8% yield. The second Suzuki–Miyaura cross-coupling at the C-3 position was then carried out on the intermediate, after iodination with pyridin-4-ylboronic acid, in the presence of Pd(PPh_3_)_4_ and Na_2_CO_3_ in a solution of 1,2-dimethoxyethane (DME)/H_2_O under microwave irradiations at 130 °C for 1 h, which supplied the product **42** with a 34% yield (Figure 30).

Pandit et al. [71] reported the synthesis of a new series of 3,6-disubstituted imidazo[1,2-*b*]pyridazine **43** and a TNF-α evaluation as the inhibitor. As reported in Figure 31, *N*-(4-(imidazo[1,2-*b*] pyridazin-6-yloxy)phenyl)acetamide was obtained in a 71% yield starting from 6-chloroimidazo[1,2-*b*]pyridazine with substituted phenol in the presence of cesium carbonate and copper iodide as the catalyst in DMF at 140 °C under microwave irradiations for 40–120 min. A reaction of *N*-(4-(imidazo[1,2-*b*] pyridazin-6-yloxy)phenyl)acetamide with *N*-bromosuccinimide at a temperature of 25–30 °C in chloroform produced the intermediate in an 86% yield. Next, the desired compounds **43** were achieved by Suzuki cross-coupling using boronic acid or its pinacol ester with the previous intermediate in the presence of Pd(PPh_3_)_2_Cl_2_ as the catalyst and potassium carbonate in DMF/water, under microwave irradiations at 140 °C for 0.5–1 h. This protocol provided the compounds **43a** and **43b** with a 45 and 65% yield, respectively (Figure 31).

## 6. Quinoline

Quinoline, also recognized as benzopyridine, 1-benzazine and benzazine, is a significant *N*-containing organic heterocyclic aromatic molecule characterized by a double-ring structure consisting of a benzene ring fused to pyridine at two adjacent carbon atoms. Quinoline represents one of the many components extracted from coal tar in 1834 and it was first synthesized in 1879. The quinoline derivatives are present in various natural species, principally in alkaloids; for example, quinine was extracted from the bark of the Cinchona tree in 1820 and was used to cure malaria. Several conventional and microwave irradiation reactions have been developed to obtain quinoline derivatives (for example, the Skraup, Doebner, Doebner Miller, Pfitzinger, Gauld Jacobs, Riehm, Combes, Conard–Limpach and Povarov reactions); moreover, some drugs containing quinoline scaffolds (such as quinine, cinchonine, amodiaquine, ciprofloxacin, topotecan, mefloquine, and pelitinib) present anti-malarial, anti-bacterial, anti-inflammatory, anti-microbial, anti-cancer, anti-tubercular, and anti-HIV activities [72].

### 6.1. Microwave-Assisted Synthesis Synthesis of Quinoline Derivatives

In a paper published in 2018 by Rani et al. [73], the high yielding microwave-assisted synthesis of 4-aminoquinoline-phthalimides **44** (Figure 32) and their anti-plasmodial activities evaluation, was reported. Important results were obtained in terms of the reaction times and yields; moreover, most of the compounds displayed good antiplasmodial activity.

New hydrazide–hydrazone motif quinoline derivatives have been reported by Ajani et al. [74]. 2-Propylquinoline-4-carbohydrazide was treated with several types of ketones, isatin and a coumarin presence in an ethanol solvent for 1–2 min, via microwave irradiation, and this led to the desired products of **45a**–**45g** (Figure 33).

A one-pot Ugi-four component condensation via microwave irradiation was applied by Thangaraj et al. [75] using aniline derivatives, 2-methoxy quinoline-3-carbaldehyde derivatives, lipoic acid and cyclohexyl isocyanide to obtain the quinoline-based peptides **46**. The products were obtained with a high purity and excellent yields (Figure 34).

In a paper published in 2019 by Mandlenkosi Robert Khumalo et al. [76], the design was reported of a simple, quick and eco-friendly, microwave-assisted multicomponent synthesis. Novel pyrazolo-[3,4-b]-quinolines **47** were obtained in aqueous ethanol as a reaction medium (Figure 35). This reaction was characterized by an operational simplicity, good selectivity, mild reaction conditions, a brief reaction time and with no byproduct formation.

Rao et al. [77] reported a simple and effective one-pot, two-step microwave-assisted synthesis: 2-nitrobenzaldehyde and indoles that reacted in the presence of the chemo selective reductant Tin(II) chloride (SnCl_2_) and ethanol as the solvent to obtain substituted 3-aminoarylquinolines **48**. This kind of reaction led to 3-aminoarylquinoline derivatives in good to moderate yields (Figure 36).

The synthesis of quinoline derivatives has been reported by Devi et al. [78]. The first step of this procedure consisted of a reaction between *o*-nitro benzaldehyde and a β-keto ester in the presence of iron powder, HCl and ethanol EtOH, under microwave irradiation for 15 min to obtain the quinoline derivative **49** (Figure 37). Subsequent reactions performed by MW allowed the preparation of hydrazide or oxadiazole derivatives.

In 2020, El-Naggar et al. [79] reported an efficient and simple procedure for the microwave-assisted synthesis of new quinoline derivatives **50**–**52**, carried out by reacting 2-chloroquinoline-3-carbaldehyde with aryl aniline and active methylene compounds (Figure 38).

The eco-friendly synthesis of benzo-fused indolizines, such as ethyl 4-bromo-1-(substituted-benzoyl) pyrrolo[1,2-*a*]quinoline-3-carboxylate analogous **53,** has been reported by Vijayakumar Uppar et al. [80]. The synthetic procedure was based on the one-pot microwave irradiation of 3-bromo quinolone, phenacyl bromide and electron-deficient alkynes in the presence of a triethylamine base and acetonitrile (Figure 39).

Li et al. [81] have reported the synthesis of polyheterocyclic-fused quinoline-2-thiones **54** starting from the ortho-heteroaryl anilines and carbon disulfide CS_2_ in water under microwave irradiation, at 140 °C for 30 min (Figure 40).

New methyl piperazinyl-quinolinyl a-aminophosphonates (MPQ-APs) **55** were obtained by Rajkoomar et al. [82] through a one-pot reaction with methyl piperazinyl-quinolinyl carbaldehyde, diethyl phosphite and aniline derivatives in the presence of Pd-SrTiO_3_ catalyst under microwave conditions (Figure 41).

In 2020 Patel et al. [83] developed the synthesis of quinoline-4-carboxylic acid derivatives **56** via a one-pot, three-component reaction from aromatic benzaldehyde, substituted aniline, and pyruvic acid, using *p*-toluenesulfonic acid (p-TSA) under microwave irradiation. This synthetic strategy has various advantages, such as a higher yield, an easy work-up process, the avoidance of harmful organic solvents and a brief reaction time (Figure 42).

Tasqeeruddin et al. developed an efficient and green synthesis of quinoline derivatives. Substituted 2-aminoaryl ketones with the active methylene compounds were reacted using L-proline under Knoevenagel condensation, affording the desired quinolone derivatives **57**. Additionally, the L-proline was found to be an efficient catalyst for the Knoevenagel condensation. The reaction (Figure 43) was carried out under conventional as well as under microwave conditions. The microwave-assisted procedure was found to be much more efficient in terms of the time and yield [84].

In 2021, Pradeep et al. [85] reported the microwave-assisted synthesis of pyrrolidinyl–quinoline-based pyrazoline products **58** via a Michael addition reaction involving pyrrolidinyl–quinoline chalcones and hydrazine hydrate (Figure 44).

A revised microwave-assisted click chemistry method was employed to synthesize the quinoline–triazole conjugates 59a–59c, using the synthetic procedure reported in Figure 45 [86]. This method offers several advantages, including higher yields, shorter reaction times, and environmentally friendly conditions.

An ultrafast technique (10 s) was used for the microwave-assisted synthesis of 2-quinolinone-fused-lactones **60** from Fenton’s reagents in formamide obtaining the desired products with satisfying yields (up to 46% 10 s) (Figure 46) [87].

In a recent paper, Bhuyan [88] reported the synthesis of quinoline derivatives 61 starting from 2-naphthylamine, aldehyde and diethyl-2-benzylidenemalonate. This synthesis was accomplished through the utilization of a microwave-assisted catalyst and a solvent-free, three-component, one-pot aza-Diels–Alder reaction technique (Figure 47).

Seleno [2, 3-b] quinoline derivatives **62** were obtained by Attia et al. [89] by using transition metal cobalt oxide nanoparticles (Co_3_O_4_ NPs) under different microwave irradiation powers and irradiation times via click chemistry (Figure 48).

A microwave-radiated condensation of 6/7/8-substituted 3-bromomethyl-2-chloroquinoline was reported by Marganakop et al. [90]. The synthetic strategy started from 2-chloro 6/7/8-substituted quinoline-3-carbaldehyde with 1,2-phenylenediamine to obtain quinoline-fused 1,4-benzodiazepines 63 (Figure 49).

Quinolin-3-ylmethyl-1,2,3-triazolyl-1,2,4-triazol-3(4H)-ones 64 were obtained by Nesaragi et al. [91] through a [3 + 2] cycloaddition of azides with terminal alkynes via click chemistry (Figure 50). This approach offers several benefits, such as rapid reaction times, an easy purification process, impressive yields ranging from 88% to 92%, the attainment of a high purity and regioselective single product synthesis through microwave irradiation.

Zhang et al. reported a protocol for the microwave-assisted Friedländer synthesis of quinoline derivatives **65a**–**65g** obtained by the condensation of 2-aminobenzophenone with the corresponding carbonyl compounds in the presence of phosphomolybdic acid and under solvent-free conditions (Figure 51). The fluorescence properties of the provided compounds were studied, and all the compounds exhibited good fluorescence characteristics. The presence of substituted groups, as well as the concentrations of quinolines and solvents, had significant effects on the fluorescence properties of the compounds [92].

The efficient MW-assisted, one-pot synthesis of new and known quinoline-3-carbonitrile derivatives **66** was achieved using novel, nano-sized cobalt–ferrate nanocomposites coated with N-doped graphene quantum dots (N-GQDs/CoFe_2_O_4_) and N-GQDs/CoFe_2_O_4_ magnetic spherical particles as a highly effective magnetic nano-catalyst. This reaction occurred rapidly, with a reaction time as short as 60–90 s. Additionally, this method offers several advantages such as its environmentally friendly approach, the ease of separating the nano-catalyst, and its ability to be reused for up to seven runs without any significant loss in the catalytic efficiency (Figure 52) [93].

### 6.2. Ultrasound-Assisted Synthesis of Quinoline Derivatives

In a paper published in 2018 by Prasad et al., the one-pot rapid synthesis of 2-substituted quinolines **67** under ultrasound irradiation in water was reported, using SnCl_2_·2H_2_O as a convenient pre-catalyst. A three-component reaction with aniline, aldehydes, and ethyl 3,3-diethoxypropionate was carried out in the presence of aerial oxygen to give the desired products in good yields (Figure 53). The obtained compounds were evaluated for their antibacterial activities, and several of them demonstrated effectiveness against both Gram-positive and Gram-negative species. Specifically, compound **67b** exhibited promising antibacterial activities against both types of bacteria [94].

## 7. Isoquinoline

Isoquinoline, a structural isomer of quinoline, is a nitrogen-containing heterocyclic aromatic compound which is composed of a benzene ring fused to a pyridine ring. As compounds containing an isoquinoline scaffold have been found to possess a wide range of biological and pharmacological activities [95,96,97,98,99,100,101], their synthesis has aroused considerable attention from the scientific community. To further explore the importance of this scaffold, new synthetic methodologies, and the modification of traditional procedures, such as the Bischler–Napieralski and Pictet–Spengler [102] and Pomeranz–Fritsch reactions [103], were developed. Among these synthetic efforts, the application of green methods, such as microwaves and ultrasound-assisted synthesis, enriched compounds containing the isoquinoline nucleus and, at the same time, reduced pollution.

### 7.1. Microwave-Assisted Synthesis of Isoquinoline Derivatives

A convenient microwave-assisted, metal-free method for the synthesis of hydroxyl-containing isoquinolines **68** was developed by Shao et al. [104]. The synthetic protocol consisted of a microwave-assisted, metal-free tandem oxidative cyclization reaction of vinyl isocyanides with alcohol tandem oxidative cyclization (Figure 54).

The present methodology offered an efficient approach for the preparation of hydroxyl-containing isoquinolines, showing an excellent functional group tolerance and was highly atom-economical.

In 2019, Deshmukh et al. developed a new method for the synthesis of isoquinolines assisted by microwave energy [105]. This strategy used polyethylene glycol (PEG) as a green, non-volatile, and biodegradable solvent. Moreover, the authors employed the homogeneous recyclable ruthenium as a catalyst, which catalyzed the annulation reactions of ketazines with alkynes via C–H/N–N bond activation to synthesize isoquinolines **69** (Figure 55).

As compared to the previously reported synthetic procedures based on conventional heating [106], this protocol represented an environmentally friendly approach due to a shortened time of reaction, a reduced time of purification and no external oxidants or transition metals being involved.

Later, Van der Eycken et al. developed an efficient palladium-catalyzed reaction under microwave irradiation for the synthesis of 4-substituted isoquinolines **70** [107]. This protocol was performed through a palladium-catalyzed reductive cyclization/ring-opening/aromatization cascade of N-propargyl oxazolidines (Figure 56).

### 7.2. Ultrasound-Assisted Synthesis of Isoquinoline Derivatives

Recently, an efficient multi-component reaction under ultrasonic irradiation conditions was developed by Sharafian et al. to synthesize pyrido [2,1a]isoquinoline derivatives **71** [108]. This synthetic strategy involved the use of phthaldehyde, methylamine, a-halo substituted carbonyls, activated acetylenic compounds and triphenylphosphine (PPh_3_) in water at room temperature to obtain the desired compounds with excellent yields in a short time frame (Figure 57); the shorter the reaction time, the easier the work-up, and the higher the yields of the product, while the green reaction conditions represent the advantages of this method.

## 8. Pyrrolopyridine

The fused pyrrole and pyridine rings are commonly known as azaindoles. According to the mutual position of nitrogen atoms on the bicyclic ring, four different isomers of azaindoles are described: 4-azaindole (1H-pyrrolo[3,2-b]pyridine), 5-azaindole (1H-pyrrolo[3,2-c]pyridine), 6-azaindole (1H-pyrrolo[2,3-c]pyridine) and 7-azaindole (1H-pyrrolo[2,3-b]pyridine) [109]. Azaindoles are structurally similar to indoles but the substitution of a CH group with a N-atom improves the aqueous solubility and the pharmacokinetic properties; therefore, this nucleus has attracted the interest of researchers worldwide [110]. Several azaindole derivatives have shown potential activity as anti-inflammatory [111] and anticancer agents [112]. In addition, they have been studied with promising results for the treatment of Alzheimer’s disease [113] and diabetes [114]. The pyrrolopyridine nucleus is also commonly found in natural active compounds [110].

Considering the high biological importance of these heterocyclic molecules, different synthetic routes, using MW and US, to produce a variety of new azaindole derivatives are reported in the literature.

### 8.1. Microwave-Assisted Synthesis of Pyrrolopyridine Derivatives

Le et al. [115] described the synthesis of 7-azaindoles (**72a**–**p**) starting from *o*-haloaromatic amine and terminal alkynes via iron-catalyzed cyclization under microwave irradiation. After the optimization of the reaction conditions, different substituted 3-iodo-pyridin-2-ylamine and alkynes were reacted, using iron(III) acetylacetonate Fe(acac)_3_ as a catalyst, under microwave heating for 60 min at 130 °C (Figure 58) to afford 7-azaindoles **72a**–**p**. Both the electron-donating group -CH_3_ and the electron-withdrawing groups -CN and -CF_3_ on the pyridine ring were tolerated in the reaction conditions. Terminal alkynes with a phenyl substituent led to a higher yield compared to aliphatic alkynes. The same procedure was applied to prepare 1,2-disubstituted 7-azaindoles **73a**–**o**, starting from 3-iodo-N-substituted pyridin-2-amine and the terminal alkyne (Figure 59).

The microwave irradiation, compared to the classical heating, was more efficient in the solvent-free synthesis of new C-3-substituted azaindoles reported by Belasri et al. [116]. The reactions were performed by using cyclic imines such as 3,4-dihydroisoquinoline; 6,7-dihydrothieno[3,2-*c*]pyridine; 3,4-dihydro-β-carboline; 4,5-dihydro-3*H*-benz[*c*]azepine and azaindoles in different conditions of reaction times and temperatures to afford, respectively, the compounds **74a**–**74d**, **75a**–**75d**, **76a**–**76d**, and **77a**–**77d** (Figure 60).

The 7-azaindoles, 4-azaindoles, and 6-azaindoles exhibited similar reactivities; however, when attempting the reaction with 5-azaindole, it was necessary to employ a catalyst of 10 mol% p-TSA. This higher catalyst amount was required due to the lower reactivity of 5-azaindole, which can be attributed to its higher pKa value (8.42) in comparison to the 6-azaindole (5.61), 4-azaindole (4.85), and 7-azaindole (3.67)

New 7-(hetero)aryl-1*H*-pyrrolo[2,3-*c*]pyridine derivatives (**78a**–**p**) were synthesized by Savitha et al. [117] via the microwave-assisted Suzuki–Miyaura cross-coupling of 7-chloro 6-azaindole and a variety of boronic acid or potassium organotrifluoroborate. The reactions were accomplished with XPhos-PdG2 as a catalyst and potassium phosphate K_3_PO_4_ as a base in DMF/EtOH/H_2_O under microwave heating at 100 °C for 30 min (Figure 61). The use of MW allowed for reducing the reaction time, avoiding the protodeboronation of boronic acid and improving the yields.

In a recent paper, Pavithra et al. [118] developed a green method to synthesize methylenated derivatives of 7-azaindoles (**79a**–**g**) under MW irradiation with an excellent yield. N-substituted-7-azaindoles, aqueous formaldehyde (37%) and montmorillonite K-10 50% *w*/*w* clay (used as a solid acid catalyst) were microwave-irradiated at 100 °C for 5 min (Figure 62). For the 3-bromo-substituted compound, the methylenation reaction took place in the C-5 position to afford bis(3-bromo-1-methyl-1*H*-pyrrolo[2,3-*b*]pyridin-5-yl) methane (**79g**).

The halogenation of different heteroaromatic compounds, including 7-azaindoles, was studied by Kour et al. [119] using tetrabutylammonium salts TBAX (X = I, Br) as the halogenating agents and oxone as the oxidant in EtOH under microwave heating for 5 min (Figure 63). Here, 5-bromo-3-iodo 7-azaindole (**80a**) and 3,5-dibromo 7-azaindole (**80b**) were obtained starting from 5-bromo 7-azaindole.

### 8.2. Ultrasound-Assisted Synthesis of Pyrrolopyridine Derivatives

Venkateshwarlu et al. [120] can be considered as the first research group to describe the ultrasound-assisted synthesis of 1,2-diaryl-substituted azaindole derivatives (**81a**–**p**). The synthetic procedure was based on a one-pot reaction via Pd/C-Cu catalysis: in the first step, *o*-bromo-substituted amino pyridine reacted with aryl iodide to form the initial C-N bond. In the second step, a C–C and a following C–N bond was established between the N-aryl-substituted intermediate and a terminal alkyne. The authors studied different reaction conditions, identifying the optimum yield that was obtained by performing the reaction with 10% Pd/C-PPh_3_-CuI as the catalysts in PEG-400 under ultrasound (Figure 64). The use of ultrasound accelerated the process, resulting in a faster and more efficient reaction.

The cytotoxicity of the synthetized compounds was evaluated against MD Anderson-Metastatic Breast-231 cells, MDA-MB-231 and MCF-7 metastatic breast cancer cell lines. The Structure–Activity–Relationship (SAR) studies showed that the position of the endocyclic “N” atom and the substituents at positions 1- and 2- influenced the activity of the compounds; 5-azaindoles resulted as being the most active agents and the derivatives with substituents characterized by electron donating groups proved to be more cytotoxic. Compound **10i**, with a p-tolyl at position 1 and a p-methoxyphenyl at position 2, showed better anti-proliferative properties than suramin, which was used as the reference in the cell-based and enzymatic assay.

## 9. Pyrrolopyridazine

Pyridazines and their fused heterocyclic derivatives or the fused pyrrolo-heterocyclic compounds, such as the pyrrolopyridazines system, are important compounds that are known for their luminescence and photochromic properties. In the meantime, these nuclei have been proven to be promising tools in medicinal chemistry; some derivatives are reported for their antimicrobial or anticancer effects such as Janus kinase (JAK) inhibitors or Mitogen-activated protein kinase (MEK) inhibitors [121]. More recently, some researchers have reported that pyrrolopyridazine derivatives were characterized by chemical moieties commonly found in PARP-1 inhibitors and, from that, they deduced that pyrrolopyridazine could be considered as a novel scaffold for PARP-1 inhibitor development [122].

### Microwave-Assisted Synthesis of Pyrrolopyridazine Derivatives

In a recent paper, Moldoveanu et al. compared the synthesis of pyrrolopyridazines by conventional thermal heating and under microwaves [123]. The general procedure for the synthesis of pyrrolopyridazine derivatives **82** and **83** is reported in Figure 65.

The preparation of the compounds **82** and **83** was based on the N-alkylation of pyridazine with bromoacetone that provided the pyridazinium bromide salt. This step was followed by a cycloaddition of pyridazinium ylides, generated in situ in the presence of triethylamine TEA with methyl propionate or dimethyl acetylene dicarboxylate. Conventional heating procedures required 6 h to give the pyrrolopyridazine derivatives **82** and **83**.

The synthesis was then carried out under MW irradiation and in a first attempt the reaction was conducted in a closed vessel (30 bars of pressure). Moldoveanu et al. discovered that the optimal reaction conditions corresponded to a temperature of 155 °C and a pressure range of 16–17 bars.

The reaction times were significantly reduced from 6 h to 10 min with a minimum consumption of organic solvents. For the compounds **82** and **83**, the yields were 90% and 91%, respectively, in comparison to 70% for the conventional heating.

In a recent paper, the synthesis of many nitrogen heterocycles obtained using a Rh(III)-catalyzed imidoyl C-H functionalization was reported [124]. The use of a Rh (III) catalyst and microwave heating allowed for the efficient preparation of pyrrolopyridazines **84** as reported in Figure 66.

## 10. Hybrid Microwave–Ultrasound Irradiation: Selected Applications

The combined use of microwaves and ultrasound is a promising solution for process intensification in physical and/or chemical processes. Combining these two sources of energy can lead to the development of a fully controllable and scalable hybrid technology that can be used in laboratories, and in pilot and industrial scales. This technology has a wide range of applications, encompassing organic synthesis, bio-waste valorization, and chemical extraction [125].

Chemists have always looked for synergism, that is, a combination of tools, reagents, or processes producing a larger effect than the sum of their individual effects. Maeda and Amemiya, among others, can be considered as pioneers of this technique as they first recognized the synergic effect of simultaneous US/MW irradiation [126].

Currently, there are no examples in the literature that describe the synthesis of the nuclei considered in our review or their functionalization. The problem probably lies in the fact that until now it has been difficult to combine the two techniques in a single reactor. As we already reported in Section 1.3 of this review, the main challenge in making a machine that is able to use a combined and/or simultaneously both sources of energy has been related to the concept of inserting ultrasound into a microwave unit or vice versa. In fact, the first examples were performed in-house [26,27].

An article, that is considered a milestone for the developing interest in the combined use of microwaves and ultrasound in the field of organic chemistry and medicinal chemistry, was published in 2001 on Green Chemistry. It reports the hydrazinolysis of esters and the preparation of hydrazides via simultaneous MW and US irradiation [127]. Different methyl esters were compared and among these, methyl salicylate was used as a model system for the identification of optimized conditions using different methods.

The isolated yield for a conventional reflux, US reaction, MW-assisted reaction, and their combinations are reported in Table 7. In addition to increasing the yield, under ideal circumstances this technology fusion enabled a significant decrease of the reaction time from 9 h to 40 s. This result was attributed to a combination of an induced mass transfer at the phase interfaces caused by US activation and MW irradiation.

Presently the interest is growing, and new reactors are available on the market. Therefore, considering that organic synthesis will undergo increased automation in the near future, this will be supported by the construction of continuous-flow systems capable of rapid, efficient, and scalable, automated processes.

There are several examples of successful synthetic procedures performed by hybrid microwave–ultrasound irradiation in the literature. Among them, it is noteworthy to mention a paper on the synthesis of aripiprazole [128]. The authors focused on improving the synthesis of the drug aripiprazole by using a simultaneous microwave–ultrasound irradiation (SMUI) technique. They used a chloro-based carbostyril intermediate (CBQ) and piperazine (DCP·HCl) moiety and they performed alkylation in the presence of a mixture of a green medium and a base. The researchers established an optimal process recipe and used a numerical tool called a response surface methodology artificial neural network (RSM-ANN) to examine and optimize the process.

The findings of the study led to several important conclusions. Firstly, the SMUI technique effectively enhanced the synthesis of aripiprazole. The combined effects of microwave and ultrasound irradiation overcame heat and mass transfer barriers, while significantly reducing the processing time compared to the traditional approach. The SMUI technique reduced the total impurity profile by approximately 45% and maintained a consistent process yield, thus minimizing the need for auxiliary purification processes.

Mokariya et al. developed a simultaneous ultrasound and microwave (US/MW)-enhanced, proficient, robust, and sustainable, three-component region-selective synthetic protocol for 3-formylindole clubbed 1,2,3-triazole derivatives [129].

The results showed that by condensing the transformations into a single step and simplifying the procedure using a small quantity of solvents, the ultrasound and microwave irradiation sped up the reaction, thereby demonstrating their great yielding potential.

The reaction times were reduced from hours to minutes; by employing this procedure, the laborious process of isolating and handling unstable organic azide derivatives is circumvented, resulting in a high yield of pure product that does not need to be further purified. The pivotal stage in this synthesis is the in situ generation of organic azide, which is subsequently fused with an alkyne derivative under the catalytic influence of copper iodide, leading to the production of regioselective 1,4-disubstituted 1,2,3-triazole.

For this synthetic protocol, no additional pretreatments, ligand additions, amine additions, or base additions were necessary.

## 11. Conclusions

In this review, we have outlined how the use of ultrasound and microwave irradiation, either separately or together, has determined advantageous outcomes for heterocycle synthesis and creative methods for green chemistry. Due to their numerous benefits, such as changing the reactivity, improving yields and selectivity, reducing the reaction time, limiting energy consumption and waste production, using water/PEG as a solvent instead of volatile organic solvents or solventless reactions, and activating catalysts, to name a few, the use of ultrasound and/or microwave is, in fact, completely in accordance with the principles of green chemistry/engineering.

This review will help provide sufficient and updated information on the different methodologies for the synthesis of heterocycles nuclei which will be useful for the derivatization of the moiety and their applications in wide areas by the scientific community.

The articles that have appeared in the literature in the last decade attest to the promise of this study for creating new, eco-friendly procedures.

The promise of these novel technologies, which can transform traditional chemistry by changing reactivities, energy consumption, and scientific paradigms, must be extensively publicized.

The aim of this compilation is threefold: firstly, it provides global coverage of the latest developments and prospects in green chemistry as they pertain to heterocyclic synthesis; secondly, it showcases cutting-edge research in the realm of bioactive heterocyclic compounds; and thirdly, it explores the potential of combining novel applications with other modern methods in organic synthesis.

To comprehend the chemistry involved and to make it easier to compare each study presented in the literature, it is vital to rigorously characterize the parameters from US and MW. It is crucial to remember that all the precise characteristics and experimental conditions reported (such as the frequency, powers, ultrasonic intensity, radical production, form and geometry of the employed reactors, the method of irradiation, etc.) must be meticulously reported in the experimental section of publications.

Additionally, a thorough comparison with equivalent silence conditions (i.e., blank reactions) is needed to demonstrate the effects of MW or US at certain temperatures; the optimized measurements with appropriate devices are crucial components for the validity of the claims.

The review highlights the researchers’ efforts in adopting simpler synthetic schemes, often utilizing one-pot reactions. Additionally, they frequently employ environmentally friendly solvents such as ionic liquids and water, as well as catalysts. Notably, some researchers have prepared the catalyst using green methods. It is worth noting that some researchers have adopted methods to generate toxic reagents in situ to prevent their manipulation and potential hazards.

The selected *N*-containing bicyclic heterocycles are the core structure in many bioactive compounds and various commercially available drugs. We like, in this regard, how some researchers have applied a synthetic green protocol that they developed for the one-pot synthesis of functionalized 2-oxo-benzo[1,4]oxazines in water using ultrasound irradiation, for the first time, the synthesis of cephalandole, an indole antitumor alkaloid. This means that we can synthesize purer drugs; thus, avoiding by-products that often cause side effects. Biologically-active molecules obtained through green methods will likely be purer and offer a safer probability for a first-in-human application; consequently, these advancements are expected to bring about important progress in enhancing human health.

Due to the continued importance of this core field in the scientific world, their synthesis needs to be seriously investigated. Medicinal chemistry and drug synthesis are the research fields where these techniques can contribute beneficial advantages and we are optimistic that we will soon move toward a new and interesting hybrid technology, that is adaptable to any scale.

We have strong confidence that this publication will provide significant benefits to both newcomers and experts in the field, serving as a catalyst for further advancements and breakthroughs in academia and industry.

## Data Availability

Not applicable.

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
