# Peer review of "The Application of Microwaves, Ultrasounds, and Their Combination in the Synthesis of Nitrogen-Containing Bicyclic Heterocycles"

_ijms, 2023, doi:10.3390/ijms241310722_

Round 1

Reviewer 1 Report

The authors enumerate a wide variety of bicyclic heterocycles synthesized using microwave, ultrasound, and their combination. This extensive list of examples highlights the applicability of these methods. The paper is well-written, but it could be enhanced by providing a more comprehensive introduction and conclusion. Currently, it only consists of reaction schemes and tables that record the reaction conditions. Readers are anticipating finding more detailed information.

1.       For instance, the paper could benefit from including a schematic about the experimental setup for the microwave and ultrasound part. Such an illustration could provide valuable detail in both sections 1 and 10. As the saying goes, "a picture is worth a thousand words”. Readers can have a visual understanding of how the experiments are conducted from the beginning. Moreover, this would simplify the understanding of the challenging and intriguing aspects of combining these two techniques, which are described in section 10.

2.       Likewise, some visual illustrations about the heating effects of microwaves and the cavitation effects of ultrasound could be valuable. The reason is that the manuscript currently lacks a comprehensive explanation of the principles underlying the utilization of microwaves and ultrasound for synthesis. For example, while the authors mention the use of 2.45GHz microwaves, they do not explain the rationale for choosing this specific frequency. This explanation is actually simple but crucial as it is linked to the selection of suitable solvents for microwave-assisted synthesis. Not all solvents can be effectively heated by microwave. Similarly, the ultrasound section does not provide a clear understanding of the cavitation effect, nor does it explain how the formation and collapse of these microbubbles facilitate the synthesis process. Providing experimental setups and results would make these principles easier to understand for readers. The current introduction part skips over many important details and there is plenty of room for the authors to go more in-depth and highlight key aspects.

Author Response

Reviewer 1

The authors enumerate a wide variety of bicyclic heterocycles synthesized using microwave, ultrasound, and their combination. This extensive list of examples highlights the applicability of these methods. The paper is well-written, but it could be enhanced by providing a more comprehensive introduction and conclusion. Currently, it only consists of reaction schemes and tables that record the reaction conditions. Readers are anticipating finding more detailed information.

Thank you for the prompt review of our submitted manuscript.

  1. For instance, the paper could benefit from including a schematic about the experimental setup for the microwave and ultrasound part. Such an illustration could provide valuable detail in both sections 1 and 10. As the saying goes, "a picture is worth a thousand words”. Readers can have a visual understanding of how the experiments are conducted from the beginning. Moreover, this would simplify the understanding of the challenging and intriguing aspects of combining these two techniques, which are described in section 10.

We thank the Reviewer for his/her helpful suggestion. We have added a schematic illustration (see Figure 1a-b, introductory section) that may simplify readers' understanding, highlighting the advantages of applying MW and US techniques for the synthesis of bicyclic heterocycles of our interest.

  1. Likewise, some visual illustrations about the heating effects of microwaves and the cavitation effects of ultrasound could be valuable. The reason is that the manuscript currently lacks a comprehensive explanation of the principles underlying the utilization of microwaves and ultrasound for synthesis. For example, while the authors mention the use of 2.45GHz microwaves, they do not explain the rationale for choosing this specific frequency. This explanation is actually simple but crucial as it is linked to the selection of suitable solvents for microwave-assisted synthesis. Not all solvents can be effectively heated by microwave. Similarly, the ultrasound section does not provide a clear understanding of the cavitation effect, nor does it explain how the formation and collapse of these microbubbles facilitate the synthesis process. Providing experimental setups and results would make these principles easier to understand for readers. The current introduction part skips over many important details and there is plenty of room for the authors to go more in-depth and highlight key aspects.

We thank the Reviewer for his/her comment. We have added a comprehensive explanation of the principles behind the use of MW and US for synthesis (lines 119-134, 145-165, 204-209), accordingly. Details on the choice of using specific frequency values and particular solvents in the microwave technique have been included in the introductory section. Similarly, in the section on the ultrasonic technique, details on the cavitational effect have been added.

Reviewer 2 Report

This manuscript presents the application of microwaves, ultrasounds, and their combinations in the synthesis of various nitrogen-containing bicyclic heterocycles that have been reported during the last several years. This article can be very useful to synthesis chemistry researchers, especially in pharmacy. However, I noticed some shortcomings that need to be taken care of to make the article very useful to the readers.

In the introduction, briefly describe how different solvents can help achieve the desired results in microwave-assisted synthesis. For example, ionic liquids, alcohols, and polyols have special advantages considering their high loss tangent. Even the combination of different solvents can enhance the reaction yield. Please discuss such features briefly.

Define abbreviations: JAK or MEK. Do so for all abbreviations in the manuscript. Define abbreviation before use.

Since this is a review, authors should express their own opinions (advantages and disadvantages) about the results they have discussed in the manuscript. The article is otherwise merely a compilation of findings from several reports.

Moderate changes required.

Author Response

Reviewer 2

This manuscript presents the application of microwaves, ultrasounds, and their combinations in the synthesis of various nitrogen-containing bicyclic heterocycles that have been reported during the last several years. This article can be very useful to synthesis chemistry researchers, especially in pharmacy. However, I noticed some shortcomings that need to be taken care of to make the article very useful to the readers.

Thank you for the prompt review of our submitted manuscript. We hope that our revisions are deemed appropriate to your feedback and that the revised paper can now be accepted in its current form.

In the introduction, briefly describe how different solvents can help achieve the desired results in microwave-assisted synthesis. For example, ionic liquids, alcohols, and polyols have special advantages considering their high loss tangent. Even the combination of different solvents can enhance the reaction yield. Please discuss such features briefly.

We thank the Reviewer for his/her suggestion. We have expanded the introductory section by discussing the features suggested by the Reviewer.

Define abbreviations: JAK or MEK. Do so for all abbreviations in the manuscript. Define abbreviation before use.

We thank the Reviewer for your suggestion. We have defined all abbreviations in the entire manuscript accordingly.

Since this is a review, authors should express their own opinions (advantages and disadvantages) about the results they have discussed in the manuscript. The article is otherwise merely a compilation of findings from several reports.

We thank the Reviewer for your helpful suggestion. We have completely rewritten the conclusion section to express our views.

Comments on the Quality of English Language

Moderate changes required.

We have improved the quality of English language as requested.

Round 2

Reviewer 2 Report

The amended changes are acceptable and therefore, I recommend accepting the manuscript for publication.